# Access Control Design Practice and Solutions in Cloud-Native Architecture: A Systematic Mapping Study

**DOI:** 10.3390/s23073413

**Published:** 2023-03-24

**Authors:** Md Shahidur Rahaman, Sadia Nasrin Tisha, Eunjee Song, Tomas Cerny

**Affiliations:** Department of Computer Science, ECS, Baylor University, Waco, TX 76798, USA

**Keywords:** security, access control, cloud-native, microservice, authentication, authorization

## Abstract

Protecting the resources of a cloud-native application is essential to meet an organization’s security goals. Cloud-native applications manage thousands of user requests, and an organization must employ a proper access control mechanism. However, unfortunately, developers sometimes grumble when designing and enforcing access decisions for a gigantic scalable application. It is sometimes complicated to choose the potential access control model for the system. Cloud-native software architecture has become an integral part of the industry to manage and maintain customer needs. A microservice is a combination of small independent services that might have hundreds of parts, where the developers must protect the individual services. An efficient access control model can defend the respective services and consistency. This study intends to comprehensively analyze the current access control mechanism and techniques utilized in cloud-native architecture. For this, we present a systematic mapping study that extracts current approaches, categorizes access control patterns, and provides developers guidance to meet security principles. In addition, we have gathered 234 essential articles, of which 29 have been chosen as primary studies. Our comprehensive analysis will guide practitioners to identify proper access control mechanisms applicable to ensuring security goals in cloud-native architectures.

## 1. Introduction

Cloud-native architecture refers to a set of practices and technologies that enable the development and deployment of applications in the cloud. It is an approach that emphasizes using microservices, containers, and container orchestration platforms such as Kubernetes. Cloud-native architecture is designed to take advantage of the cloud’s scalability, flexibility, and agility. It allows applications to be built and deployed faster and with more excellent reliability than traditional approaches, and it also enables organizations to better manage their resources and reduce costs. Overall, cloud-native architecture is a way of building and deploying optimized applications for the cloud environment [1]. Microservices are a fundamental component of cloud-native architecture, as they give the developers advantages to meet the challenges and growing demand from the customer end, which the monolithic applications sometimes fail to execute. Dr. Peter Rogers first introduced the topic of microweb services at a cloud computing conference in 2005 [2]. Later, in 2011, the developer community of Netflix and Amazon presented the concept of microservice architecture [3]. The purpose of a microservice is to decompose the application logic into fine-grained components with clear, coordinated boundaries of responsibility. As each element is deployed separately, microservices in a gigantic codebase address the breakdown of the complex tasks in a monolithic architecture. The flexibility, robustness, and scalability provided by the microservice architecture give the industry more incredible infrastructure to think extensively and quickly about a problem. The microservice design employs HTTP and JSON for user-to-system connections and internal service communications [4]. This rising architectural approach improves the maintenance, scalability, availability, and resilience of complex systems and introduces new challenges, especially security [5,6]. IoT-based applications have adopted security measures such as encryption, access control, authentication, and regular software updates to prevent cyber attacks and ensure the integrity, confidentiality, and availability of data transmitted and processed by IoT devices. Multi-access edge computing fueled the deployment of IoT-based applications where security concerns, especially authentication and accountability issues, are tackled with a prominent approach [7]. A monolithic program will have an average of 39 vulnerabilities for every 100 KLOC, but a microservice application would have an average of 180 vulnerabilities [8].

As the system may have a large number of services and it can acquire thousands or millions of users, it is imperative to secure communication. Moreover, security concerns evolve with the emerging number of responsibilities. For this reason, the system needs to identify authentic users who can acquire the intended resources they request. This verification is necessary because the system needs users to obtain only the resources for which they are authorized for a specific time. If an unwanted user somehow manages to circumvent a particular service’s security, they can access all of the resources. As the services inside the architecture are internally connected, the scenario can quickly make the system vulnerable and marginalized. Therefore, securing the microservice-based application with a proper access control model is essential. Access control is a crucial component of every company’s security framework. Every security system aspires to implement reduced concession, zero-trust, separation of roles, and other best practices without impairing business operations. An access management system can be organized in a variety of ways. The developers often find it challenging to choose which access control model they can implement as the implementation depends on many criteria such as customer’s needs, number of services, the design patterns, the responsibilities managed by the service discovery, etc. According to Hinkley’s industrial assessment, several firms have seen significant attacks on their microservice-based systems in recent years, ranging from single services to the entire system [9]. These security issues must be addressed during a program’s design and implementation phases. Authentication and authorization are the fundamental security criteria that must be handled throughout the design process.

We identified these patterns in our study and recommended how application security architects should leverage them. Our main findings show that most authentication and authorization challenges involving microservices are related to communication between them and the complexity of implementing security in each microservice, resulting in complexity both in development and in the increase in the attack surface because individual attention must be given to each microservice. Furthermore, our study identified those challenges and discussed the potential access control model currently implemented in the industry. Finally, our study provides a comprehensive analysis for developers to efficiently select the proper access control model and prospects of the security analysis in a cloud-native architecture.

This study does a systematic mapping analysis to identify the access control method employed in cloud-native systems. It thoroughly identifies existing research on static analysis of the defensive strategy to safeguard cloud-native systems, classifying potential access control design patterns, authentication, authorization techniques, tools, and procedures addressing the abovementioned mechanisms and managing the present difficulties and gaps. The contributions of the research are, in brief, as follows:Categorize the access control mechanism addressed in microservices and cloud-native architecture.Classify the authentication and authorization techniques in microservices and cloud-native architecture.Study the possible access control mechanism’s relationship to software vulnerabilities resulting from inappropriate access control.Description of the problems with MSA access control and the open challenges with a broader statement of purpose for the community of researchers and practitioners involved in that area.

The structure of this research is as follows: Section 2 describes the Related Works. Section 3 then discusses our Research Methods, which includes a summary of our approach for conducting a systematic literature review, research questions, and inclusion and exclusion criteria. The Research Findings and Results are presented in Section 4. Threats to Validity are discussed in Section 5. Finally, Section 6 covers the Discussion, then Conclusions and Future Work are in Section 7.

## 2. Related Works

This section reviews the existing literature on the access control model of microservice architecture. We searched the relevant studies in all significant scholarly databases and found many studies. An overview of the articles that have been evaluated is presented below.

In addition to extracting the existing security solutions, Pereira-Vale et al. [10] conducted a multivocal literature review. They introduced categorizations into variations of standard security mechanisms and scopes related to security settings. They referred to authentication and authorization as the most mentioned security mechanisms. They also analyzed the frequency of publications they found where the adapted security mechanism was authentication and authorization. However, the security mechanisms for microservices do not precisely depend on the access control mechanism, but it creates a vast open scope to which practitioners need to adapt. Those can address the attacks or vulnerabilities, potential tools, or mitigation strategies for those attacks. In our previous study [11], we thoroughly analyzed the effective mechanisms that can be applied to consider the protection strategy of cloud-native systems.

Hannousse and Yahiouche et al. conducted a thorough mapping analysis to highlight the critical vulnerabilities to the security of microservice-based systems [12]. They also discussed methodologies and tools used to investigate and validate the solutions provided and security mechanisms utilized to recognize, mitigate, and prevent such risks. Although it covered some of the same ground as our study, it placed more emphasis on risks and classifies them differently than we do.

Trnka et al. [13] systematically presented an analysis to discover the Internet of Things security solutions. They provide insight into the standard approaches and benchmarks applied for authentication and authorization. Although their research focused on authentication and authorization, the semantic procedure of microservices or cloud-native architecture still needs to be answered. De Aguiar Monteiro et al. [14] surveyed security, privacy, and standardization in cloud computing environments for microservices’ architectures. They identified four critical protection features as their primary conclusions: containers, data, permissions, and network. Additionally, they provide a set of protections against security risks, including host-authenticated TLS with in-band authentication, mutual transport layer security, and principal propagation through security tokens. The prime difference between their document survey and ours was how they focused on the development perspective of the microservice-based system, which did not shed light on several aspects in detail.

A strategy for distributed services was proposed by Rudrabhatla et al. [15], who also discussed how specific security measures employed in software solutions for microservices need to be revised. However, no effective technique for microservice authentication and authorization was suggested in this study.

A comprehensive literature review was performed by Almeida et al. [16] focusing on the issues, solutions, and tools related to authentication and authorization in microservices. They identified that the literature concentrated on identifying the open-source solutions of the specifications. However, considering the relative topics and findings, this study needs to comprehensively analyze which access control mechanism we can integrate.

Ponce et al. [17] presented a multivocal literature review for identifying ten bad smells indicating security issues in microservices. They showed that the issue of insufficient access control appears if a microservice-based application fails to implement access control in one or more of its microservices, potentially breaching the confidentiality of the data and business operations of the microservices where access control is missing. They also provide a mitigation strategy for this issue by implementing OAuth2.0., and Open Authorization (OAuth) 2.0, an established security protocol. The difference between their analysis and ours is that we comprehensively focused on systematically mapping the current research areas. Finally, our analysis covered the full scope of research with thorough details, merging the smells and the issues of access control mechanisms, and the security aspects of cloud-native systems. Similar to the previous study, Soldani et al. [18] considered the access control mechanisms one of the pains of the microservice-based systems. They conducted a systematic grey literature review to address the pains and gains of microservice-based systems. Their investigation led to defining access control as one of the pains of designing concern of the development. Finally, they presented a mitigation strategy by delivering studies on addressing the pain by providing tools supporting a consistent, decentralized access control.

The studies mentioned above have addressed work on microservices relevant to the procedures and techniques of developing systematic mapping studies in microservices outlining security challenges, authentication, and authorization, following the combined study themes in Table 1. However, significant research has yet to analyze the potential pattern of access control in a microservice architecture.

## 3. Research Methodology

This section outlines our in-depth review of the protocol and customized tactics used in the systematic mapping study. We begin by identifying the research questions, looking up relevant papers in the literature using various data sources, personally vetting the automatically selected publications to weed out those irrelevant to our study, and then snowballing. Then, we examined these papers to compile statistical and clear responses to our study questions.

### 3.1. Research Questions

Microservices have brought significant security issues regarding the gigantic architectural structure. The prime attack surface for the microservice is the API gateway, which the developers and security analysts consider the policy enforcement point [19]. The access control mechanism should be defined, and the pattern the system should follow must validate this point. Practitioners need to understand the basic functionalities and approaches considering the architectural bindings and security issues that might arise due to the improper access control mechanism. Thus, we must define the challenges and consider the potential strategies we can apply to adopt the justifiable access control pattern in cloud-native systems. We aim to shed light on those strategies and provide awareness of comprehensive access control methods to the developers that can be used as an asset to microservice development and design. Therefore, the research objectives we can adapt are as follows:Identify and classify attacks or vulnerabilities resulting from improper implementation of access control mechanisms in cloud-native architectures.Determine the access control strategy to defend against attacks.Determine the appropriate authentication and authorization mechanism which can contribute to maintaining the security life goals of cloud-native systems.Identify existing tools or approaches implemented for access control.Discover the shortcomings found and concentrate on the strategies and challenges for each objective mentioned above.

### 3.2. Research Questions

RQ1:What are the most common access control mechanisms that are practitioner utilized in cloud-native architecture?aWhat classification or taxonomy does each of these techniques fit into?RQ2:What authentication and authorization techniques are addressed in the current literature?aWhat is the taxonomy or classification of these tactics?RQ3:What tools or approaches addressing the techniques align with proper access control mechanisms in the literature?aWhat functionalities are supported by them?RQ4:What are the existing limitations and challenges in the cloud-native architecture’s access control implementation?

We offer some explanations below to support our decision to consider the above research questions.

**RQ1:** The first research question aims to provide an overview of current access control mechanisms used in cloud-native architecture. This question is foundational, as it provides a basis for understanding the current state of access control mechanisms in cloud-native architecture.**RQ2:** The second research question builds on the first, exploring the factors influencing the selection and implementation of authentication/authorization techniques in cloud-native architecture. Authentication and authorization techniques are a subset of access control mechanisms. Access control encompasses a broad range of security measures that control access to resources, including authentication and authorization.**RQ3:** The third research question takes a more technical perspective, exploring the integration of access control and authentication/authorization with other cloud-native technologies and the challenges of managing these mechanisms effectively in complex, multi-cloud environments. This question is important because it considers the practical implications of implementing access control and authentication/authorization in cloud-native architecture.**RQ4:** The fourth research question is more forward-looking, focusing on the future research challenges and opportunities in access control and authentication/authorization in cloud-native architecture. This question is important because it highlights potential areas for future research, such as developing new tools and frameworks to support the implementation of these mechanisms, and exploring new approaches to optimizing their performance, scalability, and security.

We believe we highlight the structured approach taken toward exploring a focused area of research. The foundation of the research area is established, and then specific components or subsets of the concepts are analyzed in detail. The practical implementation of these concepts is also addressed, highlighting their relevance in real-world scenarios. Furthermore, we end the analysis by providing insight into the potential areas of future research. By categorizing and identifying these areas, it is clear that this research is ongoing and that there is a desire to continue to explore and improve upon these concepts.

### 3.3. Searching Methodology

We had to choose a sufficiently broad selection query to collect relevant research papers. The primary research was retrieved through a search of five major digital libraries. The following is a list of the databases:ACM Digital Library;IEEE Xplorer;Springer Link;Scopus;Science Direct.

The search queries we used to search the above databases are given in Listing.

**Listing 1.** Search Query.
("access control" OR "access right" OR "authentication"

OR "authorization" OR "identity")
AND
("design pattern")
AND
("micro service" OR "micro-service"
OR "cloud-native" OR "cloud native")


### 3.4. Study Selection

Before collecting papers, two filtering stages were produced by the automated search. In the first phase, titles and abstracts were examined to determine relevance. The second stage involved looking at the full texts of publications to see whether they complied with our inclusion criteria. Then, using the works cited in the already selected articles, we employed snowballing to locate more relevant sources for our research. After the first two steps, each reference obtained this way underwent screening. All referred publications were included in the dataset for the selected papers, and these actions were approved. For these recently added articles, snowballing was applied repeatedly until it reached a fixed point or until no more documents were added to the dataset.

### 3.5. Inclusion Criteria

When strict inclusion and exclusion criteria are applied, the number of publications in online academic libraries falls. Only peer-reviewed publications from journals and conferences are included in this study. We define the inclusion criteria as follows:Publications published since 2012;Articles written in English;Publications including studies conducted with access control design patterns of microservice architectures as their primary topics;Articles addressing security solutions covering the authentication and authorization mechanism;Publications with full text available in the specified databases;Research publications recommending strategies, frameworks, methodologies, or tools to deal with the prevalent access control pattern used by cloud-native systems.

#### Exclusion Criteria

Articles that have not undergone peer review;Available research papers in the chosen databases without the full text;Publication addressing only the development and design aspects of cloud-native architecture;Editorials and tutorial papers;Publications that outline the overall architectural concept but do not mention access control in microservices;Papers published as short paper (fewer than 3 pages).

### 3.6. Data Extraction and Synthesis

We gathered and encoded the pertinent data from each primary study after choosing them from the academic literature. We began by removing the metadata, which had entries for the title, publication year, source, and type of publishing. In addition, we performed document scanning in response to each of the research questions we created, taking into account the microservice access control design pattern, authentication, authorization techniques, and already-used solutions.

## 4. Result

This section will outline the mapping studies’ results and offer a thorough response to the research questions we have established.

### 4.1. Result Analysis

We searched in February 2023 and eliminated any publications from before 2012. The number of articles we discovered is shown in Table 2, which summarizes our article search results from the five web databases we used. We initially gathered 234 research publications using our search terms. Then, we used our inclusion and exclusion criteria to remove 172 items from consideration. The study’s consistency with prior findings was then determined after looking at the titles and abstracts of 19 papers. We gathered 43 papers and reviewed them to choose their applicability in light of the study’s objectives, and 15 were excluded by reading the full text. Afterward, we removed two duplicate articles, added three more documents, and used the snowballing procedure for the selected study. Eventually, we considered 29 research papers as the proposed main references for the study. Figure 1 shows the selection process. In addition, we analyzed the year-wise research trend in the databases by finding the number of articles in those databases in Figure 2. A detailed analysis of the chosen studies is presented in Table 3.

### 4.2. Access Control and Design Patterns in Cloud-Native Systems: RQ1

Cloud-native systems’ access control traditionally depends on using the strategies from practitioners. They focused mainly on the design pattern related to development, deployment, secure development, access control, and the corresponding performance evaluation. The classification is illustrated in Table 4, In Figure 3, we show how the category is distributed and go into further depth after that.

**Design pattern related to deployment:** The deployment of an internal cloud service, positioned behind an internal load balancer that spawns new service instances following load requirements and registers them with the load balancer, resulting in seamless scalability and elasticity is part of Kogias et al.’s [27] focus on the practice of pattern. Concerning removing the load balancer from the critical channel, they provide direct communication latencies. For example, the proposed system CRAB, a Connection Redirect Load Balancer, enables traffic redirection to handle complex load balancing policies. This approach’s advantage is managing the workload for the load balancer, which includes user authentication and gives the service communication flexibility. However, the web tier and the backend layer can scale separately and remain independent due to utilizing the two load balancers in this design pattern.**Design pattern related to development:** Vale et al. [37] presented the rationality of the design pattern practitioners employed in developing microservice-based systems. They also provided the pains and gains in those patterns and evaluated the quality attributes by tracking key facets of software quality. Finally, they asserted that practitioners only sometimes use QAs as the word to express and assess the properties of their systems. They divided the design patterns into seven categories, where security is one of them. The practitioner’s survey indicates that gateway routing is considered a gain among those patterns, providing a single access layer decoupling between the client and the services. Waseem et al. [48] conducted a survey and responses from practitioners and found six results, including business capability, domain-driven, API gateways, and backend, where the most common MSA patterns are an API gateway and backend for the frontend. They provided answers to the problems that microservice-based systems’ security issues raised due to unauthorized access to the system. They recommended combining DevSecOps [49] with microservices to handle security issues and adopt modern security solutions from cloud service providers. Carranza-García et al. [46] pointed out some of the key UML language components that should be customized or enhanced to make it easier to create microservice-based software for ubiquitous systems. These UML features are identified using the case study scenario of an intelligent approach to verify attendance in a ubiquitous learning environment. Sebrechts et al. [39] described a sophisticated disconnection of cloud-native architecture and fog computing and proposed a set of design patterns and a fog-native architecture to utilize the fog fully. Applications are built using intent-based workflow building, consisting of loosely coupled microservices chosen to meet user needs best. Microservice grouping under a single proxy, frictionless user-microservice and inter-microservice connections, and request aggregation are all made possible by a new fog mesh. The design presents a unique softwarized fog mesh that supports end-user aggregation, external communication, and inter-microservice connection. Secure API communications are crucial in microservice development, considering the gigantic architecture. Platform-agnostic design patterns or standard remote API technologies such as RESTful HTTP and Web services (WSDL/SOAP) are described by Zimmermann et al. [23]. They introduced attribute-based access control management to address security issues for data privacy and integrity. They emphasized the importance of proper authentication over inter-service communication with API keys.**Secure multi-tenancy cloud pattern:** A multi-tenant framework that adds sufficient multi-tenant functionality to Kubernetes was suggested by Zheng et al. [36]. They employed role-based access control (RBAC) to reduce excessive tenant resource usage. To offer acceptable security, though, more is required. They demonstrated how separating the control plane for resource sharing significantly improves API compatibility and security management with the proper access management with tenants. The focus of Adewojo et al. [38] is on the cloud’s multi-tenancy issue, namely the tenant-isolated and dedicated component pattern. Their research is driven by the need to service a broad consumer base, use available resources, and benefit from economies of scale because it is cost-effective to provide three levels of multi-tenancy patterns—shared component, tenant-isolated component, and dedicated component—and enforce access rights with tenants that completely secure information and data privacy.**Secure access control management:** To ensure security for the organization, it is essential to have a proper access control mechanism. Yan et al. [45] presented the IoT business procedures, common access technologies, and the microservice architectural design idea. They explored the application of a unified access security gateway in the transmission sector. The development of the security gateway indicated the implementation of the access control center, the Identity Authentication Center, which provides secure communication management with automated routing.

### 4.3. Authentication and Authorization Mechanism in Cloud-Native: RQ2

Considerably, the solutions for authentication and authorization primarily depended on the architectural design and decision. Precisely, we found the security policy mainly employed in an API Gateway. Thus, we classified our findings based on the criteria below, and Table 5 lists the classification and corresponding references. We illustrate the distribution of the category in Figure 4 before going into further detail.

**Security assessment in an API Gateway:** The continuous security assessment is challenging while also developing and maintaining. The security team needs to monitor and evaluate the system’s security requirements and goals based on several specific criteria. However, the gigantic structure of how authentication and authorization can be employed, which can be utilized in security assessment, needs attention as it can break the confidentiality and integrity of the system. For example, Torkura et al. [20] addressed this, defined the security assessment, and provided a token-based authentication and authorization scheme for the system.**Development approach:** Haugeland et al. [30] presented the migration strategy from a monolithic architecture to microservice-based systems. They suggested the second stage of the migration includes implementing an API gateway and back-end communication. The inclusion of an identity server handles the different features to provide significant security by managing authentication and authorization for the various tenants. In addition, it demonstrated how the TenantManager supplies the services with endpoints for tenant-specific modifications while the API gateway acts on the user’s behalf to obtain access to the resources and oversee user login, authentication, and permission. Jayawardana et al. [35] provided a creative method for modeling domain expertise and business processes in a single specification that might result in boilerplate code that complies with microservice architecture. They utilized an API key and JSON Web Token (JWT) token-based authentication in their framework, which provides comprehensive insight into the authentication mechanism using Transport Layer Security (TLS) and OAuth2. In addition, the framework implements the functionality corresponding to the security protocols, which is the critical pathway of this research.**Architectural considerations:** Researchers have established several strategies for microservices to adapt an authentication and authorization mechanism in their applications. Pereira-Vale et al. [47] addressed the standard security mechanism of microservice-based systems where identification, authorization, and authentication are the most noted mechanisms. The use of authentication and authorization in microservice-based architectures is based on the imitation in each microservice of the techniques used for monolithic architectures by the Authenticator and Authorizer security patterns; as a result, each service operates its database or a shared database that stores credential data, but implements its user verification. Several studies in their analysis support those strategies mentioned. Alonso et al. [44] carried out an in-depth literature review on the difficulties of multi-cloud native applications, focusing on authentication, authorization, or privacy methods to handle security-related issues. By sharing data across several providers, using multiple clouds considerably increases the benefit of data confidentiality while overcoming the four significant drawbacks of cloud computing for data storage: loss of availability, corruption of data, loss of privacy, and vendor lock-in. Jiao et al. [29] presented an analysis to address the performance monitoring and security issues with microservices. Designers and researchers have developed a cloud-native application architecture based on Kubernetes. These architecture features are present: it breaks the link between environment-related tasks such as database deployment, performance detection deployment, and service request security verification, allowing microservice applications to focus more on the execution of business logic. The microservice application uses the authorization and authentication module to manage API calls, ensuring the security of the microservices and taking the intricate nature of requests into account in the vast microservice network. To incorporate several authentication and authorization mechanisms in their application, Bánáti et al. [40] looked into several of them. Their practical use offers a crucial examination of sensitive data security solutions, particularly the problems with authentication and authorization.

### 4.4. Tools and Approaches Used in the Existing Literature: RQ3

Practitioners took several approaches and utilized tools to encounter security issues related to access control and design pattern. Table 6 lists the references that addressed the techniques and tools. Before getting into more specifics, we show how the category is distributed in Figure 5.

**Container-based access control approach:** To use the isolation advantages of having the two microservices run in distinct containers, Suneja et al. [22] presented an alternative strategy called container fusion. The utility microservice can receive the access and capabilities necessary to perform its functionality under the boundaries between the two containers defined via diffusion mechanisms.**Encryption-based approach:** Dixit et al. [33] addressed the single point of attack of cloud-based systems for health record retention that aim to protect data privacy. They developed a system that combines Semantic Web and Multi-Authority Attribute-Based Encryption (MA-ABE) technologies to offer a secure, semantically rich method for facilitating data exchange across organizations that control various end-user attributes using a uniform dataset. Their strategy implements Multi-Authority Attribute-Based Encryption, which can securely provide the privacy of electronic health records, and enforces attribute-based access control to ensure the proper access rights.**Identity management solution:** Pohn et al. [31] proposed the identity management service model framework (IMSMF), which was developed in a generic service-oriented manner with a focus on the supporting toolkit of Enterprise Architecture using the open enterprise modeling language ArchiMate, which is the most popular modeling technique. They launched AuthNZ, a business solution that handles authentication and authorization and starts the access management procedure. Several forms of authentication and permission are available through this broker. By managing trustworthy third parties (TTPs), the business cooperation third party AuthN Broker expands the core business. Google and Facebook may be among these TTPs.**Architectural consideration:** A dynamic granular access control approach was proposed on top of several authorization frameworks by Preuveneers et al. [32]. In continually growing cloud and edge microservice federations, implementing this architecture provides the delegation of management of access control decisions to many stakeholders. Moreover, with only a relative performance overhead, the approach can effectively accommodate many owners for each resource. Poniszewska-Marada [43] designed and implemented a web application based on the microservice architecture to assess the usefulness of this architecture for developing corporate online applications. How the authentication and authorization are handled with a separate authentication microservice has been demonstrated for implementing the architectural perspective. This microservice comprehensively manages access rights and performs Data Encryption Standard (DES) encryption with a token-based mechanism. Yang et al. [41] provided an architectural management solution that addressed several identity and access control issues in cloud-based systems. The comprehensive advantages of the architecture provide secure access, standardization, scalability, and encryption, which can optimize and enhance the security of the systems.

### 4.5. Challenges Practitioners Addressed: RQ4

We divided the problems researchers in the literature addressed into five groups. Table 7 lists the categorization and the relevant references. Figure 6 illustrates the distribution of the category before going into greater detail.

**Security challenges on authentication and authorization:** To address the problems of flaws, errors, malfunctions, and mistakes, Waseem et al. [21] conducted a qualitative study on 1345 issue talks taken from five open-source microservices applications. They divided the security challenges they described into categories for authentication and authorization, managing authorization headers, and shared authentication. Furthermore, they provided explanations for several errors that occurred when the JWT token was implemented where the lite rest API anticipated it to be in the authorization module. Their findings also emphasized the need for open-source microservices systems where SSL connections could not be made while using Docker 2.3.0.3 with Windows Subsystem for Linux 2 (WSL2) support to integrate security certificates and standards. In-depth analysis of the security issues and solutions commonly mentioned by microservices practitioners was presented by Billawa et al. [24]. Here, we can acquire the challenges and corresponding guidelines in the existing literature in access control, authentication, and authorization in microservices.**Challenges related to database:** Secure data communication is one of the critical challenges in microservice systems. Practitioners need to adopt several mechanisms for that. Given this information, Oceans et al. [26] detailed and discussed conventional methods of protecting databases through encryption, presenting use cases for cloud and container technology, secure remote computing with SGX, homomorphic encryption, etc., and discussing their effects on security. In this study, the authors provide an overview of database encryption and management in the cloud and microservice architecture, which give a comprehensive analysis.**Communication:** Two communication patterns, choreography and orchestration, are understood by Megargel et al. [42], who also provided a decision framework for microservices collaboration patterns that aids solution architects in articulating their objectives, contrasting the critical factors, and selecting a pattern using a weighted scoring mechanism. Using three case studies, they demonstrated which one to pick for secure communication.**Challenges related to design patterns:** A cloud architecture that ensures managing failure even if crucial components briefly malfunction was given by Bau et al. [34]. They examined three-level designs, such as Software-as-a-Service (SaaS), Platform-as-a-Service (PaaS), and Infrastructure-as-a-Service (IaaS). They showed how to integrate secure authentication, overcome identity management issues, and maintain by design robustness, scalability, and flexibility.**Challenges related to improper access control:** In their discussion of cloud add-ons, Bui et al. [25] discussed instances where attackers may deliver malicious data to cloud services using their document-sharing and messaging capabilities. They mentioned to prevent add-ons from having unrestricted access to user information in the host application, cloud application suppliers often incorporate permission-based access control. There is a list of permissions that each add-on needs to function. For example, when users launch the host program for the first time or when it is installed, they are often prompted to accept the permissions explicitly. Their research revealed that this tendency is coarsely grained, requiring the user to give all requested rights for all user data or a single document. Moreover, they think that because Cross-Site Scripting (XSS) attackers execute their malicious scripts within the add-on context, such access control is useless in guarding against them. Servos et al. [28] analyzed the attribute-based access control mechanism (ABAC) and comprehended the utilization of this in cloud computing. The introduction of a domain-centric approach to implementing web services in cloud computing applying ABAC has been investigated. The current process and several methodologies provide a general idea of how to employ this in maintaining web security.

## 5. Threats to Validity

Systematic reviews, mapping studies, and surveys frequently have several validity risks that must be addressed. For example, we have found many dangers. From the viewpoint of Wohlin’s taxonomy, we explore the validity risks in this situation.

Validity Construction: Our study’s identification of primary studies from the numerous articles retrieved in the literature poses a threat to its construct validity. We have followed the guideline from [50] to design the secletion of search engine. The study topic was the primary factor in developing the search strategy, which directed us to choose the search query. To include more research for consideration, we produced two iterations of snowballing. Lastly, to guarantee that only the best studies were included, we developed a set of strict inclusion and exclusion criteria where only articles published in peer-reviewed journals and conferences were accepted because of their quality and adequate outcomes. Since we are referring to the access control mechanisms of a cloud-native application, some of the research on IoT-based solutions have been skipped because of the intended scope.Internal validity: To retain the internal validity of our findings, we took great caution. The data extraction from the group of included research raises a problem for internal validity. We managed the selection by involving two authors, where the initial selection has been considered and then we all came to a point where we considered the inclusion of the articles. We devised a method of looking for pertinent literature using the given keywords, then applying a snowballing procedure backward to the selected publications to reduce the risks.External threats: The external validity is concerned with the usefulness of a collection of findings in a wider context. Using the collected material, the classification techniques employed in this mapping were developed. The analysis and suggestions are given in the discussion to construct the baseline for the access control mechanism focused on this investigation’s difficulties and potential future expansion.Conclusion threats: Our study focuses on using categories for access control mechanisms and authentication–authorization strategies to maintain the conclusion validity of our findings. In practice, several varieties are examined, but only some allow for accurately classifying all the identified research. Finally, we selected a categorization based on a more thorough examination of the articles and solutions in the identified articles. Some of our classification categories were used just as they were or modified to fit our study’s requirements.

## 6. Discussion

Analyzing the existing literature enabled us to identify three crucial aspects of the access control mechanism. First, the design pattern to implement these and the security pattern is also addressed. Then, we extracted the communication protocol to access the system securely. We distributed the patterns for the specific areas of implementation and addressed the purpose of the implementation in Table 8.

### 6.1. Efficient Access Control Mechanism

Typically, in security, there are four types of access control policies that security analysts follow: Discretionary Access Control (DAC), Mandatory Access Control (MAC), Role-based Access Control (RBAC), and Attribute-based Access Control (ABAC). Choosing an efficient access control mechanism is crucial due to the system’s large structure and the need to handle many user requests and responses. Both the DAC and MAC models explicitly provide the user subject the object’s access right, which has significant security issues. Most research studies that support role-based access control (RBAC) base access on users’ roles inside the system and on rules that specify which accesses are permitted for users in specific roles. Large-scale permission control applications are compatible with RBAC as a technology. RBAC is a proven access control method built on ongoing use. We need to adopt the following for the implementation of the RBAC:Setting up the UID with access to a system;Defining the role;Assessing the permission equivalent to access rights;Mapping between a user and a group of roles allocated to the user during a working period, or in a session;Defining an object, which is a system resource that needs access authorization;A protected network.

We can employ the following rules in RBAC:A user can only operate if the topic has a role assigned to it;Operations do not include identification and authentication;Procedures are used to carry out all user activities.

The prime advantage we can take from RBAC is individually permitting or canceling access by grouping individuals according to their responsibilities. Establishing a set of roles in a small or medium-sized business is easy.

### 6.2. Security Design Pattern of Cloud-Native Systems

Several security design patterns we can extract from the practitioner’s point of view.

**Protecting attack surfaces:** Numerous services that various clients and systems may access are often included in microservices applications. Since it is hard to keep track of every service, it exposes them to many security vulnerabilities. Because of this, we need to set up an API Gateway to manage, monitor over, and inspect all incoming traffic before it is forwarded to the intended service. The API Gateway is the security policy enforcement point, where we can define a set of rules, communication protocol, and service registry which can protect our internal service instances.**Defense-in-depth:** Different backend services have different priority and sensitivity levels. There cannot be a single, universal approach that applies to all services. Depending on the microservice, the protection level should differ. For example, there might be open-read APIs, for instance, accessible to all callers. Once the caller has been verified, more services may be available, including premium or administrative services that must be approved based on the user profile and rigorously guarded against unauthorized access. A straightforward STS cannot meet this demand with an API gateway. We can employ in-depth defense that provides multi-layer security controls to tackle this. There can be an additional layer of security in the form of a private API gateway with an extra filter and authorization scheme for the use case when a different level of protection needs to be established for premium or admin services. To access these secure inner layers, a caller must use façade services and complete an additional authorization step.**Applying security principle, least privilege scheme:** The least privilege concept must guide API design. Only authenticated and authorized users should be granted access to the API. Access to APIs should only be allowed when necessary. Begin with the minimum necessary access and increase it only as needed.**Service monitoring:** Proper access control mechanisms must also monitor the service instances. We need to check the health information of the service instances regularly and deregister the service if it is “down” for a certain period. A faulty or unaddressed service instance can cause severe vulnerabilities in the system.

### 6.3. Communication Mechanism

The subject of secure communication in microservices is crucial. We need to make HTTPS the default for entire applications. One should encrypt sensitive data as soon as feasible before transferring it, then decode it as late as possible. Examples of sensitive data include passwords, keys, and secrets. Sending this information in plain text is never a good idea. The two primary communication mechanisms we can infer from our findings are given below:**Mutual Transport Layer Security (mTLS):** A trustworthy certificate authority has produced a public/private key pair for each microservice. The client then authenticates using mTLS using the key pair. Each microservice is identified and established by mTLS using x.509 certificates. Each certificate is signed by a reputable certificate authority and includes a public encryption key and an identity (CA). To establish encryption keys specific to each communication, each microservice in a service mesh checks the other’s certificate using the public keys. mTLS provides a secure way to ensure that each microservice connection is verified, authorized, and encrypted as privacy compliance needs to increase and zero-trust security becomes the cornerstone of enterprise cybersecurity policies.**JSON Web Token (JWT):** A very well-liked method for user authorization in microservices is JWT (JSON Web Token). This standard provides secure communication between two parties and is used to generate access tokens for applications. In this scheme, the client receives a token developed by the authentication server and certified as belonging to the user. For each future request, the client will transmit the token back to the server, allowing the latter to identify the request’s origin. The client contacts the authorization server to obtain an encrypted access token when a request is made. User information is included in this access token, which is provided to microservices. Services may verify and decode the token to identify the user accessing it. JWT typically has a payload (information on the authorization and its expiration) and a header (information on the encryption scheme), both of which are signed with the identity service secret (HMAC) or private key (RSA).**Encrypted communication:** To obtain the most significant isolation, services can be effectively broken down utilizing various controlling principles, such as the Single Responsibility Principle (SRP) and the Common Closure Principle (CCP). However, because the transactions involve numerous companies, interservice contact is inevitable. In addition, the data are stored in several heterogeneous database technologies according to the requirements where managing data security is challenging. The security requirements for the data at rest may be satisfied by encrypting it using methods such as Vormetric transparent encryption or by utilizing secure keys that are changed once every few days. Furthermore, the storage containers for the cache systems and file storage should be encrypted using strong passwords. Data that are in transmission should be encrypted using certificates and SSL layers.

### 6.4. Future Trends and Open Issues

Cloud-native systems, designed to operate in cloud environments, have access control mechanisms and design patterns that provide security for their resources and services. We have already addressed and categorized access control, design issues, and authentication and authorization mechanisms and how we can utilize those to ensure robust security in cloud-native applications. However, there are still some limitations to these access control mechanisms and design patterns. For example, cloud-native access controls frequently enable access at the resource or service level. Still, they need more granular access control at the object or data level due to a need for granular access controls, which may result in users needing more permissions, compromising security. Then, cloud-native access controls may need to provide complete visibility into access events, such as who accessed what resource, when, and for what purpose. This process can make it difficult to detect and respond to security incidents. In addition, this system can have complex access control policies that require specialized knowledge to configure and manage, leading to configuration errors that can compromise security. Furthermore, cloud-native access controls often depend on the cloud provider’s security mechanisms and infrastructure, which can limit control and visibility over security measures. Moreover, if suitable access control procedures are not in place, this system’s capability for multi-tenancy can raise the risk of data breaches and unauthorized access.

In addition to the limitations discussed earlier, there are several open issues with cloud-native systems’ access control mechanisms and design patterns. For instance, many organizations still have legacy systems that must be integrated with cloud-native systems. However, these legacy systems’ access control mechanisms and design patterns may not be compatible with those of the cloud-native systems, leading to potential security vulnerabilities. Moreover, cloud-native systems frequently use the access control method known as Role-based Access Control (RBAC), which can be rigid and might offer more granular access control than complex situations require. Furthermore, identity and access management (IAM) is another critical component of cloud-native systems’ access control mechanisms. However, managing identities and access across multiple cloud environments can be complex and challenging, leading to potential security risks. In addition, cloud-native systems may be subject to various compliance and regulatory requirements, such as HIPAA (Health Insurance Portability and Accountability Act), PCI (Payment Card Industry), and GDPR (General Data Protection Regulation). Ensuring compliance with these requirements can be challenging, particularly regarding access control. On the other hand, these systems are still evolving, and access control mechanisms and design patterns need to be more standardized, which can lead to clarity and consistency in security practices across different cloud environments.

Future studies on access control methods and design patterns for cloud-native systems must prioritize strengthening security, manageability, and usability while addressing the unique challenges these environments provide. In addition, this system has various potential future directions, which are continually developing. The following are some possible growth areas:**Context-aware access control:** Future access control mechanisms can consider the user’s context and the resource being accessed, such as location, device type, and time of day, to make more informed access control decisions.**Fine-grained access control:** Fine-grained access control mechanisms may be developed that allow for more granular control over access to specific data objects and resources within cloud-native systems.**Zero-trust security:** Zero-trust security models may become more prevalent in cloud-native systems where access is granted on a need-to-know basis, and users are continuously authenticated and authorized based on their current context.**Artificial intelligence (AI) and machine learning (ML):** AI and ML technologies may be incorporated into access control mechanisms to enable more automated and intelligent access decisions and reduce the risk of human error.**Multi-cloud access control:** As more organizations adopt multi-cloud strategies, access control mechanisms may be developed that allow consistent access control policies across multiple cloud environments.

Finally, these future directions show that access control mechanisms will continue to evolve and become more sophisticated to address the complex security challenges cloud-native systems pose.

## 7. Conclusions

This paper presented a systemic mapping analysis of cloud-native systems’ access control mechanisms and design patterns in great detail. The study investigated 29 research articles from 234 utilizing inclusion, exclusion criteria, and snowballing. The results show that the Role-based Access Control mechanism is the most efficient access control mechanism. For secure communication to handle the client requests, we must implement Mutual Transport Layer Security (mTLS) or JWT. The security design patterns need to be considered at the architectural level to prevent the system from attacks. In our future work, we would like to implement a detection mechanism using static analysis to identify the potential attacks and vulnerabilities in service-to-service communication or in an API Gateway, which will provide in-depth analysis of the security mindset for the attackers.

## Figures and Tables

**Figure 1 sensors-23-03413-f001:**
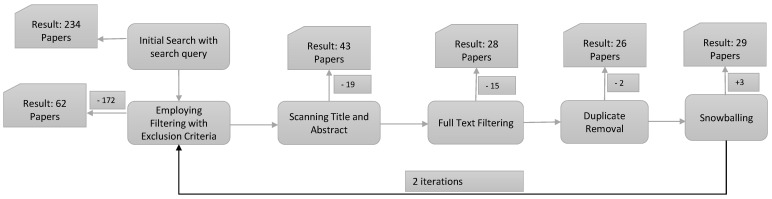
Selection procedure.

**Figure 2 sensors-23-03413-f002:**
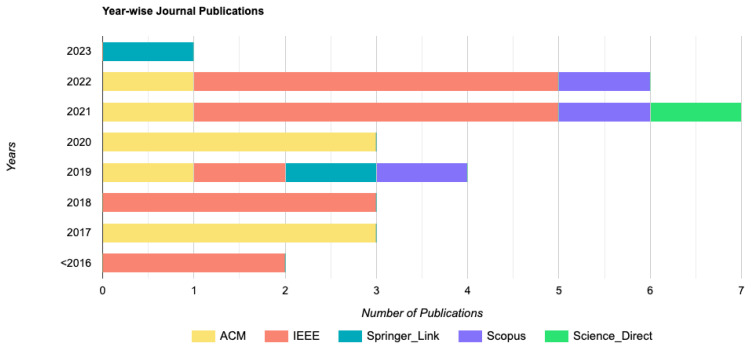
Selected studies distribution by year.

**Figure 3 sensors-23-03413-f003:**
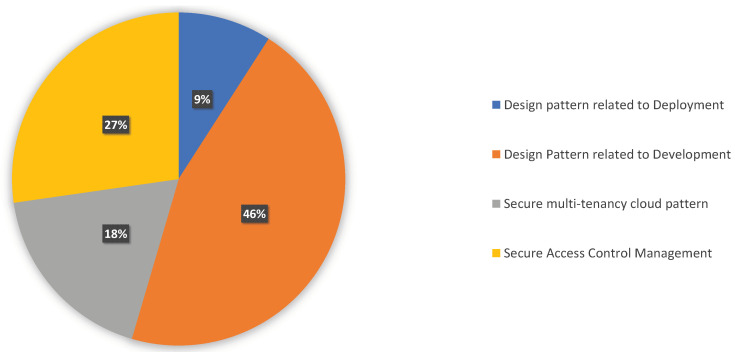
Distribution of Access Control and Design Patterns of Cloud-Native Architecture.

**Figure 4 sensors-23-03413-f004:**
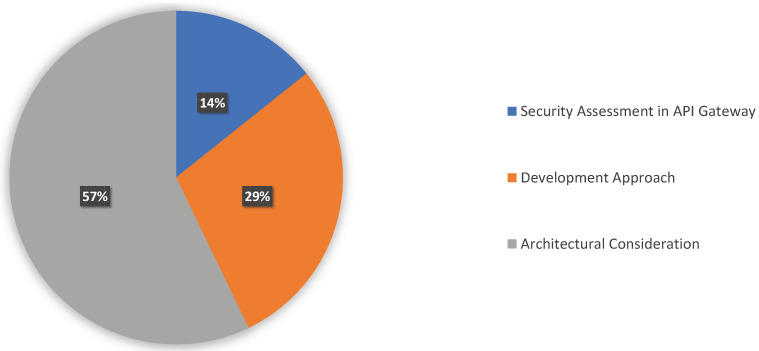
Distribution of Authentication and Authorization Pattern in Cloud-Native Architecture.

**Figure 5 sensors-23-03413-f005:**
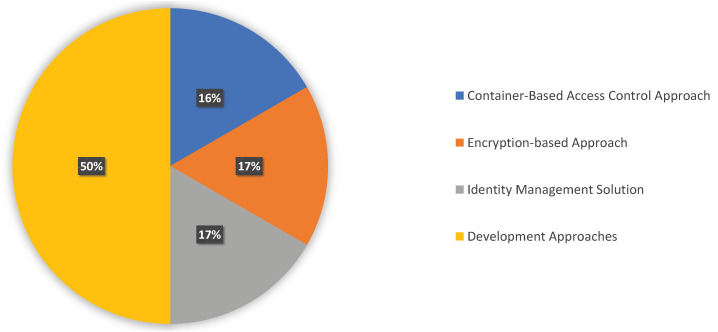
Distribution of Tools and Approaches used in the Existing Literature of Cloud-Native Architecture.

**Figure 6 sensors-23-03413-f006:**
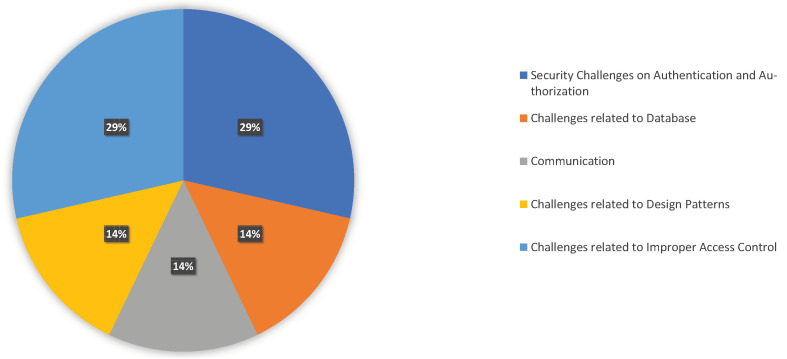
Distribution of Challenges Practitioners Addressed in Cloud-Native Architecture.

**Table 1 sensors-23-03413-t001:** Research questions of existing secondary studies in related work.

Research Questions Addressed in Related Works	
**Research Question**	**Citation**
**RQ1:** How has the frequency of publications on security in microservice-based systems varied along time? Furthermore, how have the selected publication publishers changed? **RQ2:** What research methodologies have been used to study security of microservice-based systems? **RQ3:** Security Solutions Classification **RQ3.1:** What security mechanisms have been proposed or studied in microservice-based systems? **RQ3.2:** What is the security scope of studies in microservices- based systems? **RQ4:** What security contexts have been addressed by research?	[10]
**RQ1:** What are the most-common defense mechanisms for microservices to face security-related issues based on static analysis? What is the taxonomy/categorization of these strategies? **RQ2:** What attacks and vulnerabilities are addressed by these strategies? What is the taxonomy/categorization of these attacks? **RQ3:** What tools or approaches exist in the literature? What features do they support? **RQ4:** What are the current gaps in the defense mechanism based on static analysis?	[11]
**RQ1:** What are the most addressed security threats, risks, and vulnerabilities of microservices and microservice architectures, and how they can be classified? **RQ2:** What are existing approaches and techniques used for securing microservices and microservice architectures, and how they can be classified? **RQ3:** At what level of architecture are the proposed techniques and approaches applicable for securing microservices? **RQ4:** What domains or platforms are the focus of existing solutions for securing microservices and microservice architectures? **RQ5:** What kind of evidence is given regarding the evaluation and validation of proposed approaches and techniques for securing microservices and microservice architectures?	[12]
**RQ1:** What is the taxonomy of security solutions? **RQ2:** Which topologies, communication types, and perspectives are most dominant in the authentication and authorization IoT research? **RQ3:** What are the applicability domains and requirements of identified solutions?	[13]
**RQ1:** What are the challenges mentioned in the literature to perform authentication and authorization in the context of microservice architecture systems? **RQ2:** What mechanisms are used in the literature to deal with the challenges related to authentication and authorization in a microservices architecture? **RQ3:** What are the main open-source technology solutions that implement the authentication and authorization mechanisms identified in the literature?	[16]
**RQ1:** What are the issues indicating possible security violations in microservice-based applications? **RQ2:** How can microservice-based applications mitigate the effects of security issues therein?	[17]
**RQ1:** How much evidence of microservices experimentation from industry is available online? **RQ2:** What are the technical and operational “pains” of microservices? **RQ3:** What are the technical and operational “gains” of microservices?	[18]

**Table 2 sensors-23-03413-t002:** Papers extracted from particular digital libraries.

Documents by Each Journal	
**Journal Name**	**Results**
ACM	58
IEEE	64
SpringerLink	49
Scopus	31
Science Direct	32

**Table 3 sensors-23-03413-t003:** Extracted and analyzed primary studies.

Articles Selected from Journals				
**No**	**Article Name**	**Year**	**Journal**	**Cite**
1	Integrating Continuous Security Assessments in Microservices and Cloud Native Applications	2017	*ACM*	[20]
2	On the Nature of Issues in Five Open Source Microservices Systems: An Empirical Study	2021	*ACM*	[21]
3	Can Container Fusion Be Securely Achieved?	2019	*ACM*	[22]
4	Interface Representation Patterns: Crafting and Consuming Message-Based Remote APIs	2017	*ACM*	[23]
5	SoK: Security of Microservice Applications: A Practitioners’ Perspective on Challenges and Best Practices	2022	*ACM*	[24]
6	XSS Vulnerabilities in Cloud-Application Add-Ons	2020	*ACM*	[25]
7	Security and Encryption at Modern Databases	2020	*ACM*	[26]
8	Bypassing the load balancer without regrets	2020	*ACM*	[27]
9	Current Research and Open Problems in Attribute-Based Access Control	2017	*ACM*	[28]
10	Design of Cloud Native Application Architecture Based on Kubernetes	2021	*IEEE*	[29]
11	Migrating Monoliths to Microservices-based Customizable Multi-tenant Cloud-native Apps	2021	*IEEE*	[30]
12	Reference Service Model Framework for Identity Management	2022	*IEEE*	[31]
13	Towards Multi-party Policy-based Access Control in Federations of Cloud and Edge Microservices	2019	*IEEE*	[32]
14	Semantically Rich Access Control in Cloud EHR Systems Based on MA-ABE	2022	*IEEE*	[33]
15	A cloud-based architecture for an interoperable, resilient, and scalable C2 information system	2018	*IEEE*	[34]
16	A Full Stack Microservices Framework with Business Modelling	2018	*IEEE*	[35]
17	A Multi-Tenant Framework for Cloud Container Services	2021	*IEEE*	[36]
18	Designing Microservice Systems Using Patterns: An Empirical Study on Quality Trade-Offs	2022	*IEEE*	[37]
19	Enhanced cloud patterns: A case studyof multi-tenancy patterns	2015	*IEEE*	[38]
20	Fog Native Architecture: Intent-Based Workflows to Take Cloud Native toward the Edge	2022	*IEEE*	[39]
21	Authentication and authorization orchestrator for microservice-based software architectures	2018	*IEEE*	[40]
22	An identity and access management architecture in cloud	2014	*IEEE*	[41]
23	Microservices Orchestration vs. Choreography: A Decision Framework	2021	*IEEE*	[42]
24	Development of Web Business Applications with the Use of Micro-services	2019	*Springer-Link*	[43]
25	Understanding the challenges and novel architectural models of multi-cloud native applications–a systematic literature review	2023	*Springer-Link*	[44]
26	Design and Application of Security Gateway for Transmission Line Panoramic Monitoring Platform based on Microservice Architecture	2022	*Scopus*	[45]
27	Addressing Expressiveness for a UML Microservices-Based Modeling within the Life Cycle of the Ubiquitous System Development	2021	*Scopus*	[46]
28	Security mechanisms used in microservices-based systems: A systematic mapping	2019	*Scopus*	[47]
29	Design, monitoring, and testing of microservices systems: The practitioners’ perspective	2021	*Science-Direct*	[48]

**Table 4 sensors-23-03413-t004:** Access Control and Design Pattern of Cloud-Native.

Pattern Methodology	References
Design patterns related to deployment	[27]
Design patterns related to development	[23,37,39,46,48]
Secure multi-tenancy cloud pattern	[36,38]
Secure access control management	[45]

**Table 5 sensors-23-03413-t005:** Authentication and Authorization Pattern in Cloud-Native.

Pattern Methodology	References
Security Assessment in API Gateway	[20]
Development Approach	[30,35]
Architectural Consideration	[29,40,44,47]

**Table 6 sensors-23-03413-t006:** Tools and Approaches used in Existing Literature in Cloud-Native.

Pattern Methodology	References
Container-Based Access Control Approach	[22]
Encryption-based Approach	[33]
Identity Management Solution	[31]
Development Approaches	[32,41,43]

**Table 7 sensors-23-03413-t007:** Challenges Practitioners Addressed in Cloud-Native.

Methodology	References
Security Challenges on Authentication and Authorization	[21,24]
Challenges related to Database	[26]
Communication	[42]
Challenges related to Design Patterns	[34]
Challenges related to Improper Access Control	[25,28]

**Table 8 sensors-23-03413-t008:** Specific areas for Access Control and Design Patterns of Cloud-Native.

Specific Area of implementation	Pattern Name	Purpose of the Implementation
Efficient access control Mechanism	Role-based Access Control (RBAC)	access control, Authentication and Authorization
Security Design Patterns Implementation	Attack Surface Protection	access control, Authentication and Authorization
Security Design Patterns Implementation	Defense-in-depth	Authorization
Security Design Patterns Implementation	Applying Security Principle: Least Privilege Scheme	access control
Communication	Mutual Transport Layer Security (mTLS)	Microservice-Communication
Communication	JSON Web Token (JWT)	Microservice-Communication
Communication	Encrypted Communication	Data-Security

## Data Availability

Data sharing not applicable.

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
