# Peer review of "Access Control Design Practice and Solutions in Cloud-Native Architecture: A Systematic Mapping Study"

_sensors, 2023, doi:10.3390/s23073413_

Round 1

Reviewer 1 Report

This paper has presented a systematic mapping study on the access control pattern and solutions in cloud-native architecture, in particular, 29 of 234 essential articles have been chosen as primary studies.

There are some comments:
1.What is cloud-native architecture? Please give some explanation.

2. As to RQ4,  since the papers were searched from 2012, it is not guarantee that the results satisfying the state-of-the-art 'limitations and challenges'.

3. What are the relationships between the four RQs?

4. It is suggested to add a discussion on future research trends.

Author Response

Reviewer#1, Concern # 1: What is cloud-native architecture? Please give some explanation. 

Author response: Thank you for your feedback. We acknowledged including the definition of the cloud-native architecture. In the introduction section, we have added the definition and some explanation regarding cloud-native architecture from lines 18 to 25.

Reviewer#1, Concern # 2: As to RQ4, since the papers were searched from 2012, it is not guarantee that the results satisfying the state-of-the-art 'limitations and challenges'.

Author response:  Thank you for your note. The context of Microservices got broad awareness since 2014. Here are google trends. 

There are 753 results from google scholar search for “microservice” for 2004-2012; there are ​​17,400 results for 2012-now. As more studies, 2012 is a good enough start. Moreover, it is 10 years from now.

Reviewer#1, Concern # 3: What are the relationships between the four RQs?

Author response:  Thank you for your comment. We feel the necessity to provide insight into the foundation of establishing the research questions and provide a relationship among those in subsection 3.2 from lines 202 to 231.

Reviewer#1, Concern # 4: It is suggested to add a discussion on future research trends.

Author response:  Thank you for your feedback. We value your comment and include future research trends in our discussion section 6.4.

Reviewer 2 Report

The paper entitled "Access Control Pattern and Solutions in Cloud-Native Architecture: A Systematic Mapping Study" is in general well written while it deals with an interesting research topic which is micro-services.

However, I have some minor comments prior publication:

1) Please pay attention to some syntax and grammar errors. For example, in the introductory section I read that "An study of the possible access control"

2) I do not understand why specific details regarding selection criteria for papers should be provided in the paper. These rules are in general well known and acceptable by the vast majority of researchers.

3) There are additional works in cloud native systems for IoT applications, such as the following one: Trakadas, P.; Nomikos, N.; Michailidis, E.T.; Zahariadis, T.; Facca, F.M.; Breitgand, D.; Rizou, S.; Masip, X.; Gkonis, P. Hybrid Clouds for Data-Intensive, 5G-Enabled IoT Applications: An Overview, Key Issues and Relevant Architecture. Sensors 2019, 19, 3591. https://doi.org/10.3390/s19163591

4) Although the authors have successfully extracted the most relevant questions regarding security considerations in cloud native applications, the inclusion of potential future research directions and open issues will leverage the added value of the paper.

Author Response

Manuscript ID: Sensors-2286863

Reviewer#2, Concern # 1:  Please pay attention to some syntax and grammar errors. For example, in the introductory section I read that "An study of the possible access control"

Author response: We appreciate this comment, and we followed your suggestions. We addressed the grammatical issues and proofread the text to make it consistence.

Reviewer#2, Concern # 2:  I do not understand why specific details regarding selection criteria for papers should be provided in the paper. These rules are in general well known and acceptable by the vast majority of researchers.

Author response:  We followed systematic literate review protocol by Kitchenham. It is recommends to share all these information to support reproducibility.

Reviewer#2, Concern # 3:  There are additional works in cloud native systems for IoT applications, such as the following one: Trakadas, P.; Nomikos, N.; Michailidis, E.T.; Zahariadis, T.; Facca, F.M.; Breitgand, D.; Rizou, S.; Masip, X.; Gkonis, P. Hybrid Clouds for Data-Intensive, 5G-Enabled IoT Applications: An Overview, Key Issues and Relevant Architecture. Sensors 2019, 19, 3591. https://doi.org/10.3390/s19163591

Author response:  Thank you for suggesting the research article. This research article provides the current state of hybrid clouds and other pertinent areas, such as anomaly detection, anonymization, and serverless data lakes. The importance of the study as our scope and include this paper in our Introduction section from lines 40 to 45.

Reviewer#2, Concern # 4:  Although the authors have successfully extracted the most relevant questions regarding security considerations in cloud native applications, the inclusion of potential future research directions and open issues will leverage the added value of the paper.

Author response:  Thank you for your feedback. We feel the importance of including future research trends and open issues in our analysis. In Section 6.4, we have added the Future Trends and Open Issues for the access control mechanism in the cloud-native system.
